# Buffy Coat Score as a Biomarker of Treatment Response in Neuronal Ceroid Lipofuscinosis Type 2

**DOI:** 10.3390/brainsci13020209

**Published:** 2023-01-27

**Authors:** Siyamini Sivananthan, Laura Lee, Glenn Anderson, Barbara Csanyi, Ruth Williams, Paul Gissen

**Affiliations:** 1Department of Inherited Metabolic Diseases, Great Ormond Street Hospital, London WC1N 1EH, UK; 2Institute for Health Research Great Ormond Street Hospital Biomedical Research Centre, University College London, London WC1N 1EH, UK; 3Department of Children’s Neurosciences, Evelina London Children’s Hospital, London SE1 7EH, UK

**Keywords:** neuronal ceroid lipofuscinosis type 2 (CLN2) disease, lysosomal storage disorder, intracerebroventricular, enzyme replacement therapy, disease progression, neurodegeneration, biomarker, blood buffy coat, electron microscopy, curvilinear inclusions

## Abstract

The introduction of intracerebroventricular (ICV) enzyme replacement therapy (ERT) for treatment of neuronal ceroid lipofuscinosis type 2 (CLN2) disease has produced dramatic improvements in disease management. However, assessments of therapeutic effect for ICV ERT are limited to clinical observational measures, namely the CLN2 Clinical Rating Scale, a subjective measure of motor and language performance. There is a need for an objective biomarker to enable assessments of disease progression and response to treatment. To address this, we investigated whether the proportion of cells with abnormal storage inclusions on electron microscopic examination of peripheral blood buffy coats could act as a biomarker of disease activity in CLN2 disease. We conducted a prospective longitudinal analysis of six patients receiving ICV ERT. We demonstrated a substantial and continuing reduction in the proportion of abnormal cells over the course of treatment, whereas symptomatic scores revealed little or no change over time. Here, we proposed the use of the proportion of cells with abnormal storage as a biomarker of response to therapy in CLN2. In the future, as more tissue-specific biomarkers are developed, the buffy coats may form part of a panel of biomarkers in order to give a more holistic view of a complex disease.

## 1. Introduction

Neuronal ceroid lipofuscinosis type 2 (CLN2) is a rare, rapidly progressive neurodegenerative lysosomal storage disorder. It is 1 of 13 different subtypes of neuronal ceroid lipofuscinoses (NCL) [1,2,3]. The reported incidence of CLN2 is 0.1–7/100,000, with variation depending on geographical region [4,5]. It has an autosomal recessive pattern of inheritance and is caused by deficient activity of the enzyme tripeptidyl peptidase 1 (TPP1) [6,7]. Loss of TPP1 enzyme activity leads to accumulation of ceroid lipofuscin [8,9], an autofluorescent lysosomal storage material [10]. This results in catastrophic neuronal degeneration throughout the CNS and retina [11,12,13].

The most common CLN2 genetic variants largely lead to a late-infantile phenotype which display a high genotype–phenotype correlation [1,14]. There are also some rarer variants that have been linked to delayed onset or prolonged disease course [15] and are described as “atypical” or “non-classical” phenotypes [16,17]. In late-infantile CLN2 disease, affected children typically have delayed language acquisition but otherwise are functionally normal until two to four years of age, and subsequently manifest various different types of seizures [18,19]. This is followed by a rapid decline in motor, language, cognitive, and visual function over a period of four to six years, with death by early adolescence [13,20]. Seizure photosensitivity detected by EEG using low-frequency photic stimulation is regarded as an early hallmark of the disease but is not often identified [21]. Many of these children are classified as having “childhood dementia” owing to the loss of previously acquired developmental skills and intellectual disability [22,23,24]. Whilst CLN2 disease predominantly affects the central nervous system, there is some evidence that the disease has important peripheral effects [25,26]. Cardiac co-morbidities, including cardiomyopathy and conduction defects, are well described in the juvenile onset form of NCL (CLN3 disease) [26] but have also been reported in atypical CLN2 [27].

Until recently, treatment of CLN2 was largely supportive [8,28], with pharmacological management of seizures and movement disorder alongside multimodal therapies including speech and language therapy, physiotherapy, and occupational therapy [5,29]. Psychosocial, family, and palliative care support is also of vital importance as the disease progresses [30]. Recently, in a trial of intracerebroventricular (ICV) cerliponase alfa (a recombinant proenzyme form of human TPP1), children with CLN2 disease showed a significant improvement in clinical course, with a less severe decline in motor and language function than in historical controls [31]. This treatment was given as a fortnightly infusion of 300 mg cerliponase alfa directly into an ICV device and was well tolerated [31]. Several subsequent independent case series of children receiving cerliponase alfa have supported these findings, concluding that the ICV treatment is safe and stabilises disease progression [32,33]. Importantly, this is the first disease modifying treatment that has been shown to be effective in any of the NCLs [31,32].

However, assessments of therapeutic effect have been largely limited to clinical observational measures, namely the CLN2 rating scale [31,34]. This rating scale is a crude measure of motor and language performance [35] and does not give any detailed information on the progress of the patients. These clinical examinations are often performed by the patients’ long-term physician and are therefore open to bias, limiting assessment of treatment efficacy on an individual level in clinical practice. Furthermore, determining the presence or absence of any therapeutic effect is typically delayed if symptoms alone are relied upon, because changes in symptoms occur relatively slowly (it is often unclear whether the absence of change is due to a slow rate of progression or because the treatment has been effective). A biomarker that reflects the underlying disease process more directly would therefore enable earlier and more objective assessment of therapeutic effect (or failure). This could act as an adjunct to clinical ratings to support timely decisions regarding modification of treatment.

From a disease characterisation perspective, one of the main consequences of aberrant accumulation of metabolic by-products is the development of abnormal cytoplasmic vacuolation of lymphocytes [36,37]. This feature can be detected by microscopic and ultrastructural examination of peripheral blood buffy coat [38,39]. A study examining 2500 blood films looking for vacuolated lymphocytes from patients with clinical indications that included developmental delay/regression, ataxia, seizures, and cardiomyopathy demonstrated that the most common pathology was NCL (typically CLN3) [36]. However, patients with CLN2 disease do not have vacuolated lymphocytes but instead have characteristic abnormal curvilinear storage inclusions [36,39]. These abnormal inclusions are present in lymphocytes and can only be visualised using electron microscopy since conventional light microscopy lacks the resolution needed to characterise such fine detail [36,40]. Visualisation of curvilinear storage inclusions in peripheral blood samples has circumvented the historical need for more invasive tests such as skin and rectal biopsies and has made it easier to detect cases earlier [41]. Electron microscopy is readily available in centres providing NCL diagnosis. With appropriate setup and regular use, peripheral buffy coat electron microscopy can be applied to routinely obtained peripheral blood samples.

Classic CLN2 disease demonstrates pure membrane-bound curvilinear profiles, which are a hallmark of the disease [36,42,43]. This specific pattern is not found in any other NCL disorder [36,40,41]. Those with atypical CLN2 disease often have a later onset of symptoms [16,17] and tend to have a mixed pattern of curvilinear storage material and fingerprint stacks seen on electron microscopy of their lymphocytes [40,44,45]. Whilst the presence of these storage inclusions is a useful diagnostic marker for CLN2 disease [36,39], to date, the inclusions have not been evaluated in detail following ERT treatment. We hypothesised that the percentage of curvilinear storage material in peripheral blood buffy coats would reduce after treatment and continue to do so over the course of treatment. Here, we investigate the change in appearance of the peripheral blood storage material over time whilst on treatment. Our main objective is to identify whether the amount of curvilinear storage material reduces over the course of treatment and could therefore act as a biomarker of therapeutic response in patients with CLN2 disease.

## 2. Methods

A retrospective cohort study of CLN2 patients receiving cerliponase alfa on compassionate grounds at our centre was undertaken. All included patients were given cerliponase alfa as part of an expanded access scheme and underwent yearly collection of peripheral blood samples (which were used for buffy coat testing in addition to routine monitoring) as part of standard care. These samples were taken prior to the ICV infusion on an annual basis. The collection of samples for this study has appropriate regulatory approvals (13/LO/0168; IRAS ID 95005; London-Bloomsbury Research Ethics Committee) and all participants provided informed written consent to participate.

### 2.1. Study Patients

We included all patients referred to a single tertiary centre who (i) had a diagnosis of CLN2 disease based on genetic testing and (ii) underwent ICV ERT treatment through an expanded access scheme. Exclusion criteria were (i) any contraindication to neurosurgery and (ii) known hypersensitivity to any component of the study drug.

To enable administration of cerliponase alpha, a ventricular reservoir was surgically implanted in all patients, with the reservoir placed under the scalp and the catheter placed in the cerebral lateral ventricle, with placement confirmed using brain magnetic resonance imaging (MRI). Cerliponase alfa was administered as an infusion of 300 mg every 14 days [31,46,47]. Antihistamine and paracetamol were administered orally approximately 30 min before each infusion.

### 2.2. Electron Micrographs

We investigated the percentage of cells containing curvilinear storage inclusions in the peripheral blood buffy coat layer (a concentrated layer of leucocytes and platelets found at the erythrocyte–plasma junction, following centrifugation) [39,48,49,50,51] and in skin cells [52]. A sample of EDTA blood was collected from each patient alongside routine disease surveillance bloods. In two patients, a skin biopsy was performed two years after initiation of treatment. Blood buffy coat and skin biopsy samples were fixed in 2.5% glutaraldehyde solution in 0.1 M cacodylate buffer at pH 7.2, followed by secondary fixation in 1% osmium tetroxide [51]. Samples were dehydrated in graded ethanol transferred to an intermediate reagent, propylene oxide, and then infiltrated and embedded in Agar 100 resin. Polymerisation was undertaken at 60 °C for 48 h. Semithin sections were stained with toluidine blue to identify appropriate areas. Moreover, 90 nm ultrathin sections were cut using a Diatome diamond knife on a Leica ultramicrotome. Sections were transferred to copper grids and contrasted with uranyl acetate and lead citrate. The sections were examined using a JEOL 1400 transmission electron microscope [36,39]. One hundred cells were evaluated per sample and any lymphocyte identified was used for analysis [40].

We recorded the percentage of abnormal cells on electron micrographs of the buffy coat prior to starting treatment and at yearly intervals after commencing treatment. To determine the effect of therapy in different cell types other than those observed in peripheral blood, two patients also underwent skin biopsies at two years post-treatment, and we recorded the percentage of abnormal skin cells on electron micrographs. These results were reported by the lead clinical histopathologist at our centre.

### 2.3. Clinical Assessment

Treatment outcomes were recorded using the motor and language domains of the CLN2 Clinical Rating Scale [31].

For all patients, age at diagnosis (defined as date of confirmatory enzyme assay), genotype, treatment start date, and adverse events whilst on treatment were recorded. We documented basic haematology and biochemistry (renal and liver function), ECG, EEG, and MRI brain results. We recorded the time until first unreversed two-point decline in the score on the CLN2 Clinical Rating Scale measuring motor and language skills or until the attainment of a combined motor–language score of 0. The performance was assessed over a period of 3 years during which each patient received a fortnightly 300 mg dose of cerliponase alfa.

The clinical rating scale was performed by the lead medical practitioner, a consultant specialist in inherited metabolic disorders. Data regarding adverse events and concomitant medications were reported at every visit. The attribution of adverse events was determined as part of standard care. The baseline measurement was the last observation preceding the first administration of cerliponse alfa. The reports from any MRI head scans that the patients had undergone (requested as part of standard care) were summarised descriptively.

### 2.4. Statistics

Buffy coat data were analysed using repeated measures one-way analysis of various (ANOVA) with Dunnett’s multiple comparison test [53], which treated the pre-treatment data as a baseline and the mean compared the mean of each subsequent data point against this baseline. Data from the final year were excluded from the ANOVA analysis as data were only available for two of the patients. Nonparametric data (e.g., CLN2 scores) were compared using the Friedman test [54], again comparing each post-treatment column with the pre-treatment column as a baseline.

## 3. Results

### 3.1. Demographics

Patient demographics are summarised in Table 1. A total of 6 patients received ERT as part of the compassionate use programme (2M:4F), with a median age at diagnosis of 4 years 2 months (2 years 2 months–13 years 3 months) and age at treatment initiation of 4 years 7 months (3 years 11 months–14 years 11 months). One patient (patient C) had an attenuated phenotype and was thus at the upper end of the range at diagnosis and treatment initiation (13 years 3 months and 14 years 11 months, respectively). The mean (SD) duration of treatment with cerliponase alfa was 2 years 5 months (3 months). Three patients were homozygous for a common mutation, one patient had one common and one uncommon allele, and two patients had two uncommon alleles. All the patients had a combined score of 1 to 5 on the motor and language domains of the CLN2 Clinical Rating Scale (the motor and language domains each have a 0–3 point subscale, with 0 representing no function and 3 representing normal function). The sum of the motor and language scores therefore range from 0 to 6 [18,55]. Two patients had a language score of 0 at the start of treatment, while the remaining four patients had a language score of at least 1. All patients had a motor score of 1 or above at the start of treatment.

### 3.2. Buffy Coats

Examples of electron micrograph images of peripheral blood buffy coats before and after treatment, in a single patient, are shown in Figure 1. These demonstrate clusters of membrane-bound, regularly arranged curvilinear inclusions, typical of CLN2 disease, taken prior to treatment initiation (pre-ERT). Following ERT, there are poorly defined curvilinear-like storage inclusions with prominent lipid droplets and electron dense inclusions.

The relationship between time since treatment initiation and the percentage of abnormal cells in the peripheral blood buffy coats is shown in Figure 2. Overall, a substantial reduction in the percentage of cells with abnormal storage was observed. The mean (95% CI) differences in percentage of abnormal cells in the buffy coats compared to the baseline measurements were as follows: one year: −3.5 (−10.8 to +3.8, *p* = 0.36), two years: −6.5 (−3.5 to −9.5, *p* = 0.002), and three years: −7.5 (−3.3 to −11.8, *p* = 0.005).

### 3.3. Clinical Outcomes

The relationship between time since treatment initiation and the CLN2 scores is shown in Figure 3. The median (range) CLN2 score at the baseline was 2.5 (1–5). Two of the six patients underwent reductions in CLN2 scores with treatment (one declining from 2 to 0 and the other from 5 to 4); the other four patients remained stable. There was no significant difference in CLN2 scores over the time points in the study (*p* = 0.25). This was in line with the findings of the clinical trial, which demonstrated stabilisation of CLN2 scores in patients treated with cerliponase alfa compared to natural historical controls [31].

Full blood count and renal and liver function remained stable throughout the study period. ECG and EEG results were grossly unchanged. Brain MRI showed reduced volume in the first year, with more stable appearances in the second year post-treatment.

### 3.4. Skin Biopsies

For two patients (E and F), skin biopsies were performed (two years after starting treatment) in order to assess response to treatment in various cell populations other than those present in peripheral blood. These samples were taken two years after treatment had commenced. The effect of treatment was different in different skin cells. Patient E (who underwent a 50% reduction in peripheral buffy coat storage material in the first year after treatment) had profuse deposits of curvilinear inclusions in secretory epithelial cells from eccrine sweat glands, with a similar (indistinct and non-uniform) morphological profile to the patient’s post-treatment buffy coat inclusions (Figure 4 and Figure 5). In patient F (who showed no change in percentage of buffy coat inclusions in the first year after treatment), curvilinear storage inclusions were present in the sweat glands but were more distinct in appearance.

There was no indication of storage in fibroblasts or endothelial cells in either of the two patients.

## 4. Discussion

In this study, we found a significant reduction in abnormal storage material seen on electron micrographs of peripheral blood buffy coats following regular ERT over 3 years, supporting our hypothesis of the biological validity of this as a marker of cellular response to treatment.

Our data suggest that the percentage of abnormal storage inclusions in CLN2 disease could be of value as an objective, visual marker of therapeutic response and a biomarker of disease activity. By comparison, the CLN2 clinical score is much better suited to the assessment of functional loss rather than any gain or normalisation of function.

We found that patients receiving ICV cerliponase alfa for CLN2 disease underwent little to no deterioration in symptoms over a three-year period, providing further evidence for the efficacy of this treatment. Our data are consistent with those recently reported [31,33]. Additionally, these outcomes represent a significant improvement when compared to historical untreated patient controls. An observational cohort study of CLN2 patients showed a rapid annual decline of 1.81 score units, with the combined motor–language scores declining from normal (score of 6) to no function (score of 0) in approximately 30 months [20]. The treatment was well tolerated within this group with no terminations, providing further evidence for the safety of this therapy.

There has been concern that ICV-delivered therapy would not have any effect on peripheral tissues. The demonstration of a reduction in the percentage of abnormal cells in the peripheral blood is significant because it suggests that therapy delivered directly to the CNS does also have peripheral organ effects in addition to previously documented neurological improvement. The exact mechanism is unclear, but it is possible that passive transfer of the enzyme from the CSF into the systemic circulation underlies this effect.

Ultrastructural examination of peripheral blood buffy coats is a reliable, minimally invasive test used for the diagnosis of CLN2 disease in childhood and has been shown to be particularly useful with harder-to-diagnose NCL variants [39]. The use of buffy coat analysis to determine response to treatment has not previously been reported.

The reduction in storage material present in peripheral blood white cells, despite the fact that these cells replenish after approximately 20 days, supports the notion that there is an excess of storage material in bone marrow progenitor cells (and/or in other reservoirs elsewhere) which takes several years to be cleared fully.

There are reports of cardiac manifestations of NCL disease, affecting conduction pathways or the myocardium itself and leading to development of a cardiomyopathy [22,24]. Pathological investigation of these structures in NCL disorders which have a high incidence of cardiac disease, such as CLN3, has demonstrated an accumulation of ceroid lipopigment [53,54,56,57]. Although the patients in our cohort did not, to our knowledge, have overt cardiac manifestations, the reduction of inclusions in both peripheral blood and skin biopsies raises the possibility that ICV therapy can have peripherally beneficial effects which are detectable by buffy coat analysis, offering a potential means to monitor non-neurological manifestations of the disease in the future.

Although only available in two patients, the skin biopsies provide a useful mechanism to correlate the peripheral blood buffy coat results with those in other cell types. The similarity in morphology of the inclusions observed in the buffy coats and in skin biopsies suggests that the buffy coats can provide a reliable insight into abnormalities in a range of cell types. However, our results also suggest that the effect of treatment may have been heterogenous across tissues, with some tissues responding better to ERT. For example, the lack of abnormal storage in endothelial cells suggests that these tissues may have responded better than sweat glands. This may reflect differences in enzyme availability between tissues. This has further implications regarding the non-neuronal effect of ICV therapy, namely the effect on cardiac pathology that is associated with CLN2 disease.

One major limitation of this study is its small sample size. However, this is a rare disease, and it is therefore difficult to obtain large cohorts for such studies. No contemporary control group is available for comparison as the ERT was delivered as part of a compassionate use program. Similarly, we were not able to obtain quantitative measurements of brain volume as this was a retrospective study and the necessary sequences were not consistently available. To address this and to increase the cohort size, we plan to collect this data routinely as part of standard care in the future.

The future of biomarker discovery arguably lies in creating a panel of biomarkers, which work synergistically to have better combined accuracy and capture more aspects of the phenotype [58]. It is increasingly clear that a single biomarker is not enough to capture and monitor both neurological and peripheral disease processes when dealing with complex neuro-metabolic diseases. A wide spectrum of technology may help to achieve this goal and could enable evaluation of disease at various scales. For example, whole organ pathology may be monitored using magnetic resonance imaging (MRI) with the use of quantitative brain MRI [59] and cardiac MRI [60]. Cellular pathology can be investigated and monitored using electron microscopy [39,40,51], whilst abnormalities in small molecules and specific proteins can be evaluated using mass spectrometry [58,61].

Approximately five new patients are diagnosed with CLN2 in the UK every year and become part of the ever-increasing cohort for the prospective follow-up study, which will include assessment of the peripheral blood buffy coat levels. This biomarker will be correlated with the continuous assessments of the CNS and peripheral organ manifestations of the disease. Clearly, buffy coat levels provide limited information on the individual organ damage and can only be used as a proxy measure. Recent publications have suggested that neurofilament light (NFL) chain can be a possible biomarker of axonal disease response in CLN2 [62,63]. All CLN2 patients with the classical form of CLN2 disease had high plasma and cerebrospinal fluid (CSF) NFL levels, which normalised after 2 years of treatment with ERT. The search for better biomarkers of central and peripheral CLN2 disease response to ERT is still in progress and we anticipate the use of a biomarker panel that will be representative of different cell- and tissue-specific responses.

## 5. Conclusions

We show that the percentage of cells demonstrating curvilinear inclusions in peripheral blood buffy coats undergoes a statistically significant reduction after ERT over 3 years in patients with CLN2 disease, supporting further investigation of this measurement as a biomarker of therapeutic response. Furthermore, the demonstration of a reduction in abnormal peripheral blood cells is significant because it suggests that ERT delivered directly to the CNS has peripheral tissue effects. It is important to evaluate these peripheral effects further both in order to develop non-invasive biomarkers for assessing treatment efficacy and to understand the potential positive effect of ERT on extra-CNS organ function in the long-term. Our take-home point is that the amount of abnormal storage material in lymphocytes can act as an objective marker of response to treatment in children receiving ICV ERT for CLN2 disease.

## Figures and Tables

**Figure 1 brainsci-13-00209-f001:**
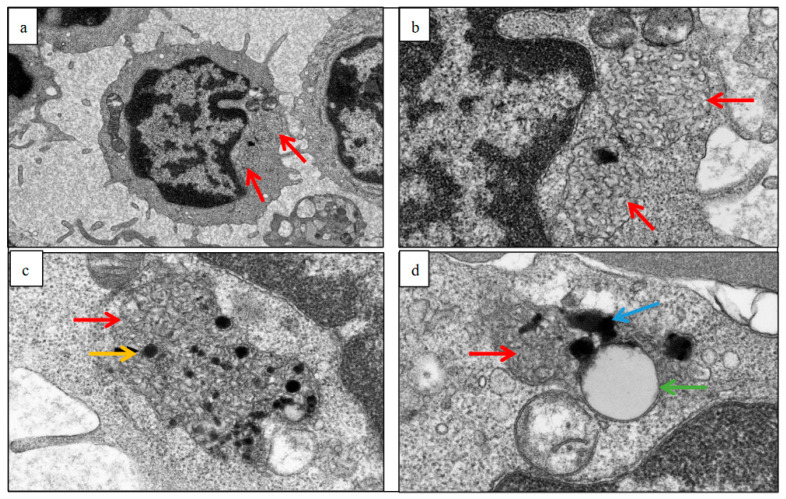
Electron micrographs of peripheral blood buffy coats pre- and post-treatment in patient E: (**a**,**b**) show low- and high-power images (mag. ×3000 and mag. ×10,000), respectively. These demonstrate clusters of membrane-bound, regularly arranged curvilinear inclusions (red arrow), typical of CLN2 disease, taken prior to treatment initiation (pre-ERT). (**c**) shows a high-power image (mag. ×12,000) one year post-ERT. There are irregular curvilinear-like inclusions (red arrow) with electron dense material representing fragments. There is also a fingerprint stack (yellow arrow) which is seen occasionally in CLN2 lymphocytes. (**d**) shows a high-power image (mag. ×15,000) two years post-ERT. There are poorly defined curvilinear-like storage inclusions (red arrow) with prominent lipid droplets (green arrow) and electron dense debris (blue arrow) (mag. ×15,000).

**Figure 2 brainsci-13-00209-f002:**
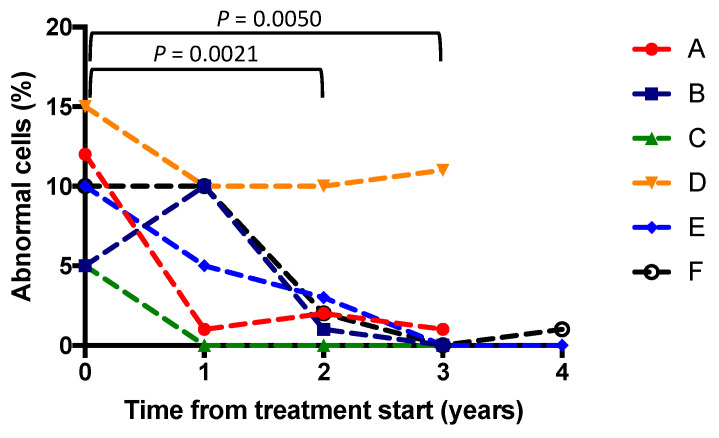
Change in percentage of cells showing abnormal storage inclusions after treatment for each of the six patients (A–F, labelled in separate colours as per the legend).

**Figure 3 brainsci-13-00209-f003:**
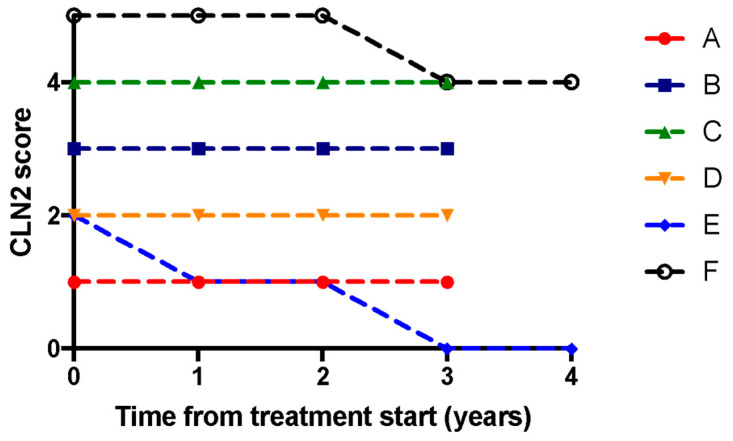
Change in CLN2 motor and language scores after treatment. Each of the six patients are represented individually (A–F, labelled in separate colours as per the legend). The dotted lines link the individual scores for each patient.

**Figure 4 brainsci-13-00209-f004:**
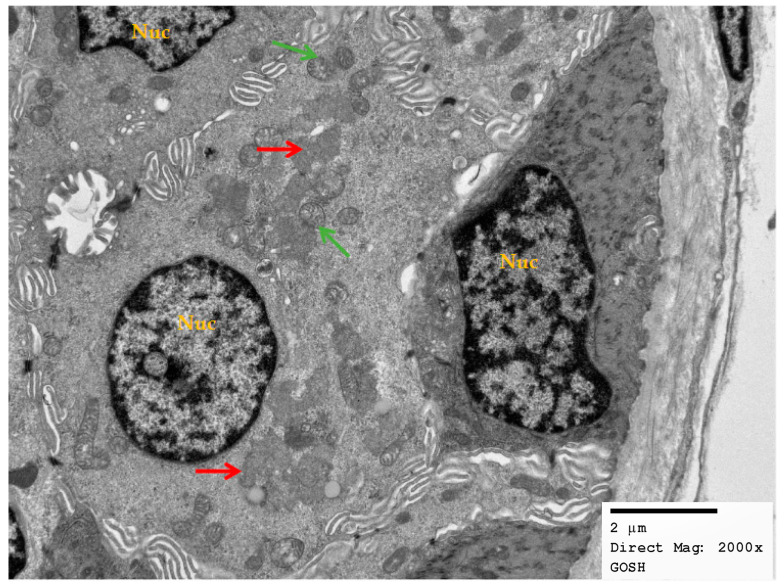
Low-power electron micrograph, mag.×2000, of secretory epithelial cells from an eccrine sweat gland with abnormal storage inclusions present in clusters of expanded lysosomes (red arrow). Numerous mitochondria with normal morphology present (green arrow) and prominent nuclei (Nuc) also with normal appearances.

**Figure 5 brainsci-13-00209-f005:**
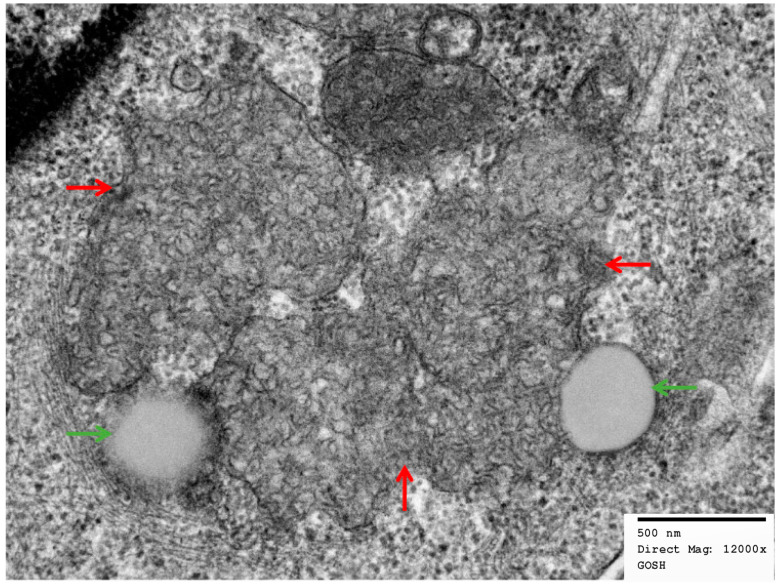
High-power electron micrograph, mag. ×12,000, of membrane-bound storage inclusions in secretory epithelial cells from eccrine sweat gland with curvilinear-like bodies (red arrow) and lipid droplets (green arrow).

**Table 1 brainsci-13-00209-t001:** Patient demographics. Six patients (A–F) are each shown in a different row.

Patient	Current Age	Age at Diagnosis	Mutation	Age at Treatment Start	CLN2 Scores at Start of Treatment (Motor, Language)
A	10 years 2 months	4 years 7 months	c.89 + 5G > A, c 509. 1G > C	4 years 10 months	1, 0
B	10 years 1 months	4 years 4 months	homozygous c.509-1G > C	4 years 7 months	1, 2
C	20 years 3 months	13 years 3 months	c.89 + 5G > C c. 1340G > A	14 years 11 months	2, 2
D	9 years 4 months	4 years 2 months	homozygous c.1052 G > T	4 years 5 months	2, 0
E	11 years 4 months	4 years	homozygous c.1052 G > T	5 years 10 months	1, 1
F	9 years 2 months	2 years 2 months	homozygous c.1052 G > T	3 years 11 months	3, 2

## Data Availability

Not applicable.

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
