# Peer review of "Buffy Coat Score as a Biomarker of Treatment Response in Neuronal Ceroid Lipofuscinosis Type 2"

_brainsci, 2023, doi:10.3390/brainsci13020209_

Round 1

Reviewer 1 Report (Previous Reviewer 1)

Thank you for addressing all the comments. After revision the manuscript looks good and can be considered for the publication.

Author Response

Reviewer 1

"Thank you for addressing all the comments. After revision the manuscript looks good and can be considered for the publication".

Thank you for your positive comments; we are pleased to see that the Reviewer recognised the value of our manuscript and was happy to proceed to publication.

 Very many thanks for your time and constructive comments, it is greatly appreciated.

Reviewer 2 Report (New Reviewer)

Introduction

1. Lines 54-55: This is the first disease modifying treatment that has been shown to be effective in children with a neurodegenerative condition [12,13]. Delete this statement as it is not accurate or necessary.

2. Lines 61-62: An objective biomarker is needed to assess response to treatment at all stages of the disease ... Change to An objective biomarker that correlates with clinical signs of disease would be helpful in assessing response to treatment at all stages of the disease ...

3. Lines 64-65: ... guide decisions regarding discontinuation of treatment. Change to ... guide decisions regarding modification of treatment.

4. Lines 75-77: These abnormal inclusions are present in lymphocytes and can only be visualised using electron microscopy since conventional light microscopy lacks the resolution needed to
characterise such fine detail [18,22 ]. The lysosomal storage bodies that accumulate in CLN2 disease have a characteristic autofluorescence. This autofluorescence in lymphocytes could be visualized in blood smears, which would be much quicker and simpler than performing electron microscopy. The authors should examine their lymphocyte samples for storage body autofluorescence to determine whether this might be a much quicker and easier method to assess storage body content of these cells.

5. Lines 78-80: It is much cheaper than molecular testing and we are able to use a routine peripheral blood sample to perform this diagnostic test. This does not appear likely to be true. Performing a TPP1 enzyme assay is quick and inexpensive and can be performed on blood samples. Electron microscopy is quite time consuming and expensive.

Methods

1. Study patients: The authors should cite one of the publications that give a more complete description of the drug treatment.

2. Blood sample collections: The authors should indicate the timing of the blood collections relative to the timing of the drug infusions.

3. Electron micrographs: The authors need to indicate the number of cells that were evaluated per sample and describe the procedure for selecting which cells to evaluate in each sample.

4. Lines 125-126: We recorded the percentage of abnormal cells on electron micrographs of the buffy coat ... Here, as well as in the abstract of the paper, the authors need to be more precise in defining what they mean by abnormal cells. Do they mean cells containing inclusions with curvilinear contents? If so, they need to state this explicitly.

Results

1. Lines 169-172: All the patients had a combined score of 1 to 5 on the motor and language domains of the CLN2 Clinical Rating Scale (which ranges from 0 to 6, with 0 representing no function and 3 representing normal function for each of the two domains) ... This wording is confusing. If the rating scale ranges from 0 to 6 and 3 represents normal function, then what do scores of 4, 5 and 6 represent? Better wording needs to be used here.

2. Line 172-173: All the patients had a combined score of 1 to 5 on the motor and language domains of the CLN2 Clinical Rating Scale (which ranges from 0 to 6, with 0 representing no function and 3 representing normal function for each of the two domains) [7,28] and a score of at least 1 in each of the two domains when the treatment was initiated. The underlined statement is not consistent with the data in Table 1.

3. Lines 193-194: ... percentage of abnormal cells ... There is no indication of what the authors are defining as abnormal cells. Since the whole study rests on which cells they define as abnormal it is imperative that a definition be given along with the number of cells per
sample that were evaluated and a description of the process by which cells were selected for inclusion in the analyses.

4. Skin biopsies: It is not apparent that these data add anything to the study since samples were obtained from two patients at only one time point. Based on the data obtained, nothing can be said about the possible effect of the treatment on storage body accumulation in the biopsy samples.

Discussion

1. The key outcome measure of the ERT treatment for CLN2 disease is its efficacy in ameliorating progression of neurological signs. Assessing storage body content in circulating lymphocytes does not appear to relevant to assessing the key treatment outcome. The data presented in this study would not appear to be a surrogate for measuring behavioral outcomes as implied by the authors.

2, As indicated in the second paragraph of the discussion, probably the most important interpretation of the findings is that the ICV administration of recombinant TPP1 appears to have an effect outside of the central nervous system. As suggested by the authors, the most likely explanation for this effect is transfer of the enzyme from the CSF into the systemic circulation. The authors should have measured TPP1 enzyme levels in the lymphocytes to evaluate this possibility.

3. Lines 262-265: There is no evidence to support this speculative statement. It should be deleted.

4. As indicated for the results section, the data on the skin biopsies should be deleted since it cannot be interpreted relative to the ERT treatments.

Conclusions

1. The authors state: We show that the percentage of cells showing curvilinear inclusions in peripheral blood buffy coats undergoes a statistically significant reduction after ERT ... If this wording is accurate, it should be used elsewhere in the paper in place of ... percentage of abnormal cells ... The validity of this statement can only be assessed with a better description of the methodology used in data acquisition.

2. The authors state that their data suggests that ERT delivered directly to the CNS is also likely to have peripheral tissue effects. There are a number of published studies that describe non- neuronal CLN2 disease pathology (e.g. cardiac pathology). These studies should be cited to
indicated the relevance of the present study to the potential that the ICV treatment could also ameliorate these other disease signs.

3. Regarding the statement: Our take home point is that the amount of abnormal storage material in lymphocytes can act as an objective marker of response to treatment in children receiving intraventricular enzyme replacement therapy for CLN2 disease. As stated earlier, the key response to treatment is prevention of neurological decline. The relevance of storage body accumulation in lymphocytes to the neurological disease is not apparent.

Author Response

Reviewer 2

Introduction

R2.1. Lines 54-55: “This is the first disease modifying treatment that has been shown to be effective in children with a neurodegenerative condition [12,13].” Delete this statement as it is not accurate or necessary.

Thank you for your comment, we have changed the statement to lines 73-75: “This is the first disease modifying treatment that has been shown to be effective in any of the neuronal ceroid lipofuscinoses (NCLs)”

R2.2. Lines 61-62: “An objective biomarker is needed to assess response to treatment at all stages of the disease ...” Change to “An objective biomarker that correlates with clinical signs
of disease would be helpful in assessing response to treatment at all stages of the disease ...”

Thank you for this point we have entirely reworked this paragraph to clarify the rationale for our study. The paragraph now reads as follows (lines 83-90):
Furthermore, determining the presence or absence of any therapeutic effect is typically delayed if symptoms alone are relied upon, because changes in symptoms occur relatively
slowly (it is often unclear whether the absence of change is due to a slow rate of progression or because the treatment has been effective). A biomarker that reflects the underlying disease process more directly would therefore enable earlier and more objective assessment of therapeutic effect (or failure). This would act as an adjunct to support timely decisions regarding modification of treatment.

R2.3. Lines 64-65: “... guide decisions regarding discontinuation of treatment.” Change to “... guide decisions regarding modification of treatment.”

Thank you, we have made this change in the text (please see the last line of the modified paragraph above; line 90).

R2.4. Lines 75-77: “These abnormal inclusions are present in lymphocytes and can only be visualised using electron microscopy since conventional light microscopy lacks the resolution needed to characterise such fine detail [18,22 ].” The lysosomal storage bodies that accumulate in CLN2 disease have a characteristic autofluorescence. This autofluorescence in
lymphocytes could be visualized in blood smears, which would be much quicker and simpler than performing electron microscopy. The authors should examine their lymphocyte samples for storage body autofluorescence to determine whether this might be a much quicker and easier method to assess storage body content of these cells.

Thank you for this interesting suggestion. This is beyond the scope of the present article but we agree that this could be a valuable avenue for future research.

R 2.5. Lines 78-80: “It is much cheaper than molecular testing and we are able to use a routine peripheralbloodsample to perform this diagnostic test.”This does not appear likely to be true. Performing a TPP1 enzyme assay is quick and inexpensive and can be performed on blood samples. Electron microscopy is quite time consuming and expensive.

Many thanks for the comment regarding the cost of electron microscopy. In centres, like our own, where electron microscopy is set up and routinely performed, we find that it is a cheaper test compared to molecular testing.

Methods

R 2.6 Study patients: The authors should cite one of the publications that give a more complete description of the drug treatment.

Thank you for your suggestion, we have cited the NEJM paper by Schulz et al 2018, which outlines the study design and gives a more complete description the drug treatment. We have also cited the animal study using a canine model, Katz et al 2014.

Schulz et al 2018 provides the short-term follow up study. The long-term follow up paper of the patients on the original trial is currently in progress, and is yet to be submitted for peer-review.

R 2.7 Blood sample collections: The authors should indicate the timing of the blood collections relative to the timing of the drug infusions.

These samples were taken prior to the infusion, along with annual review bloods which were performed as routine standard of care. We have modified the methods section to
include this information (please see lines 130-131).

R 2.8 Electron micrographs: The authors need to indicate the number of cells that were evaluated per sample and describe the procedure for selecting which cells to evaluate in each
sample.

Many thanks for your suggestion, we have updated the methods to include this information. One hundred cells were evaluated per sample, and any lymphocyte identified
was used in the analysis (l65-166).

R 2.9 Lines 125-126: “We recorded the percentage of abnormal cells on electron micrographs of the buffy coat ...” Here, as well as in the abstract of the paper, the authors need to be more precise in defining what they mean by “abnormal cells.” Do they mean cells containing inclusions with curvilinear contents? If so, they need to state this explicitly.

Thank you for highlighting this important point, we have changed the wording to be more explicit. The change is as follows: “We recorded the percentage of cells containing curvilinear storage inclusions, instead of writing “...percentage of abnormal cells”. This change is continued throughout the manuscript.

Results

R 2.10 Lines 169-172: “All the patients had a combined score of 1 to 5 on the motor and language domains of the CLN2 Clinical Rating Scale (which ranges from 0 to 6, with 0 representing no function and 3 representing normal function for each of the two domains) ...” This wording is confusing. If the rating scale ranges from 0 to 6 and 3 represents normal function, then what do scores of 4, 5 and 6 represent?

Thank you for highlighting the lack of clarity in this explanation. We have amended, as follows:
All the patients had a combined score of 1 to 5 on the motor and language domains of the CLN2 Clinical Rating Scale (the motor and language domains each have a 0-3 point subscale, with 0 representing no function and 3 representing normal function. The sum of the motor and language scores therefore range from 0 to 6 (lines 216-220).

R 2.11 Line 172-173: “All the patients had a combined score of 1 to 5 on the motor and language domains of the CLN2 Clinical Rating Scale (which ranges from 0 to 6, with 0 representing no function and 3 representing normal function for each of the two domains) [7,28] and a score of at least 1 in each of the two domains when the treatment was
initiated.” The underlined statement is not consistent with the data in Table 1.

Thank you for highlighting this to us. We have amended to read, as follows: Two patients had a language score of 0 at the start of treatment, the remaining four patients had a language score of at least 1. All patients had a motor score of 1 or above, at the start of treatment (lines 220-222).

R 2.12 Lines 193-194: “... percentage of abnormal cells ...” There is no indication of what the authors are defining as “abnormal cells.” Since the whole study rests on which cells they define as “abnormal” it is imperative that a definition be given along with the number of cells
per sample that were evaluated and a description of the process by which cells were selected for inclusion in the analyses.

Thank you for this comment, we have used the descriptive term curvilinear storage inclusions instead of the word abnormal, in order to provide greater clarity. We have amended the methods section to include information on the number of cells, per sample that were evaluated. We evaluated 100 lymphocytes per sample, any lymphocyte identified was included in the analysis.

R 2.13 Skin biopsies: It is not apparent that these data add anything to the study since samples were obtained from two patients at only one time point. Based on the data obtained, nothing can be said about the possible effect of the treatment on storage body accumulation in the biopsy samples.

We thank you for your comments. We had included the skin biopsy findings since it provided an insight into the varied effect on different cell types. We hope to pursue this work further in the future. One of the main reasons it is important surrounds longer term follow up, especially since other systemic effects such as cardiac disease associated with CLN2 is more likely to appear as time progresses. We have included this point in the introduction and discussion.

Discussion

R 2.14 The key outcome measure of the ERT treatment for CLN2 disease is its efficacy in ameliorating progression of neurological signs. Assessing storage body content in circulating lymphocytes does not appear to relevant to assessing the key treatment outcome. The data
presented in this study would not appear to be a surrogate for measuring behavioral outcomes as implied by the authors.

Thank you for your response. We completely agree that the key outcome measure of the intraventricular ERT is ameliorating progression of neurological disease. We are not looking to create a surrogate for measuring neurological outcomes, instead we propose the use of buffy coat scores as an objective adjunct to clinical assessment. The aim is to enhance objectivity in our assessment and importantly provide a way to identify changes in response to treatment earlier in the disease course than would otherwise be possible. This is important since the current clinical scale used is a crude measure of response, having a biochemical adjunct would increase overall objectivity.

R 2.15 As indicated in the second paragraph of the discussion, probably the most important interpretation of the findings is that the ICV administration of recombinant TPP1 appears to have an effect outside of the central nervous system. As suggested by the authors, the most likely explanation for this effect is “transfer of the enzyme from the CSF into the systemic circulation.” The authors should have measured TPP1 enzyme levels in the lymphocytes to evaluate this possibility.

Thank you for this helpful suggestion, we will aim to measure TPP1 enzyme levels in lymphocytes in the future, in order to evaluate the possibility of transfer of the enzyme from the CSF into systemic circulation.

R 2.16 Lines 262-265: There is no evidence to support this speculative statement. It should be deleted.

Thank you for your suggestion, we have deleted this statement.

R 2.17 As indicated for the results section, the data on the skin biopsies should be deleted since it cannot be interpreted relative to the ERT treatments.

Thank you for your suggestion. We hope to look into the effect of centrally delivered ERT on other cell types in the future. We believe that this is an important line of questioning since CLN2 has effects outside of the central nervous system (CNS), such as the myocardium and cardiac conducting system. It would be interesting to see what the effect of centrally delivered medication is, on extra-CNS pathology. Kindly refer to lines 309-318 in the discussion.

Conclusions

R 2.18 The authors state: “We show that the percentage of cells showing curvilinear inclusions in peripheral blood buffy coats undergoes a statistically significant reduction after ERT ...” If this wording is accurate, it should be used elsewhere in the paper in place of “... percentage of abnormal cells ...” The validity of this statement can only be assessed with a better description of the methodology used in data acquisition.

Thank you for highlighting this, we have removed the word ‘abnormal’ and changed it to a more accurate description, namely, “percentage of cells showing curvilinear inclusions”. We have made this clear in the methods section.

R 2.19 The authors state that their data “suggests that ERT delivered directly to the CNS is also likely to have peripheral tissue effects.” There are a number of published studies that describe non- neuronal CLN2 disease pathology (e.g. cardiac pathology). These studies should be cited to indicated the relevance of the present study to the potential that the ICV treatment could also ameliorate these other disease signs.

Thank you for this suggestion, we have included information on non-neuronal CLN2 disease pathology along with appropriate citations. We have focused on the cardiac manifestations of CLN2 disease. Please refer to lines 309-318 in the discussion.

R 2.20 Regarding the statement: “Our take home point is that the amount of abnormal storage material in lymphocytes can act as an objective marker of response to treatment in children receiving intraventricular enzyme replacement therapy for CLN2 disease.” As stated earlier, the key response to treatment is prevention of neurological decline. The relevance of storage body accumulation in lymphocytes to the neurological disease is not apparent.

Thank you for your comments. We agree that neurological decline is the key response to treatment. However the objective assessment of neurological decline is challenging in this patient group. It is useful to have a biomarker to objectively evaluate whether or not patients are responding to treatment, this could help identify treatment non-response and therefore guide decisions regarding cessation of ERT and redirection of care.

Reviewer 3 Report (New Reviewer)

Comments on brainsci-2041195 

In this manuscript, the authors investigated objective biomarkers to enable assessments of ceroid lipofuscinosis type 2 (CLN2) disease progression and response to treatment. A total of six patients were analyzed and the data demonstrated the use of the proportion of cells with abnormal storage 21 as a biomarker of response to therapy in CLN2. The manuscript was written with clear logic and the data were presented well-organized.

1.       Scale bars should be provided for images in Figures 1, 4, and 5.

2.       The reference formats should be unified. For example, in some places, it has “Anderson 2006” and “Schulz et al 2018”, while in some other places it has numbers (with or without square brackets).

Author Response

Reviewer 3
In this manuscript, the authors investigated objective biomarkers to enable assessments of ceroid lipofuscinosis type 2 (CLN2) disease progression and response to treatment. A total of six patients were analyzed and the data demonstrated the use of the proportion of cells with abnormal storage 21 as a biomarker of response to therapy in CLN2. The manuscript was written with clear logic and the data were presented well-organized.

R 3.1 Scale bars should be provided for images in Figures 1, 4, and 5.

Thank you for your suggestion, we have included scale bars for figures 4 and 5. Since figure 1 has images with 4 different magnifications, we have included detail on the magnifications of each electron micrograph instead of an error bar, so that the clarity of presentation of this composite figure is maintained.

R 3.2 The reference formats should be unified. For example, in some places, it has “Anderson 2006” and “Schulz et al 2018”, while in some other places it has numbers (with or without square brackets).

Thank you for highlighting this to us, we have looked through the manuscript and amended this accordingly, to unify the reference format. We are very grateful for your time and constructive comments.

Round 2

Reviewer 2 Report (New Reviewer)

Line 94.  Delete the phrase "...is relatively cheap and..."  The authors indicated that the electron microscopy service at their institution is inexpensive, but this is unlikely to be true at most other institutions.  This statement is not necessary and can be deleted without detracting from the paper.

Author Response

Thank you very much for your time and constructive thoughts throughout the review process. It is greatly appreciated! We have deleted the line "is relatively cheap", in reference to electron microscopy and amended the final manuscript accordingly. Very many thanks once again. 

This manuscript is a resubmission of an earlier submission. The following is a list of the peer review reports and author responses from that submission.

Round 1

Reviewer 1 Report

Sivananthan S et al has demonstrated that peripheral blood lymphocyte characterization in the puffy coat is a potential biomarker for the CLN2 disease and can be used as a promising clinical characterization during the enzyme replacement therapy. CLN diseases are mostly devastating and till date there is no curable treatment available.  There are so far 13 genetic loci involved for several forms of the diseases. Several published papers showed overlapping clinical and molecular features among the different forms of the NCLs. 

1. Did authors check the difference in the pattern/ storage materials in different forms of the diseases? I guess there might be little differences among the diseases and it's very hard to conclude any disease improvement or characterization from the buffy coat analysis.

2. Electron microscopy is not an easy/ affordable  method to use for clinical characterization of cells over the period of lifelong treatment. 

3. Peripheral cellular characterization does not reflect the brain function. Basically, most of the neurons already died before showing the clinical phenotypes. 

This work has lots of limitations and lack of practical sense. I think this study is not matured enough.

Reviewer 2 Report

The manuscript represents a very interesting study on an attempt to get biomarkers for the replacement therapy in neuronal ceroid lipofuscinosis type 2. The only idea of a treatment for this disease, which, apparently, at clinical level, delays the neurodegeneration consequence of the disease, is already great. 

However, the follow up, is very limited so far. First, it is clear that the main objective of the therapy is to slow down the neurodegeneration, and it seems it is working at some extent, since authors comment, that although there are no statistically significant differences between scores of motor and language abilities, before and after 2 and a half years of treatment, at least, the decay observed is not so strong as in untreated patients.

However, the proposal for biomarkers is not clear. First of all, authors should indicate in the introduction the characterization of the abnormal storage bodies accumulated in the lymphocytes of CLN2 patients. Maybe a reference on the subject is missing. If these abnormal inclusion bodies cannot be seen at normal microscopy, and documented in different cells on a field, is is hard to say anything about the images shown in figures 1, 4 and 5. As these are images of one single cell, is very difficult to see this, as a characteristic of the disease. On the other hand, the magnifications are different, and the abnormal structures are not highlighted by arrows, asterisks or different marks.

The abnormal curvilinear inclusions should be demonstrated in a microscopic field in bright field or fluorescence microscopy, but not electron microscopy. Perhaps, confocal technique in bright field/fluorescence at the highest magnification of optical microscopy should be better.

On the other hand, the type of blood cells shown should be characterized. Which type of blood cell: lymphocyte, neutrophils, basophils, moncytes, eosinofils, platelets. All these cells are obtained in a buffy coat. And these cells should be able to be observed by optic microscopy, without going to electron microscopy. In electron microscopy, the group vision is lost. 

Moreover, the study needs a negative control: a field with blood cells from a healthy donors, and the study of how many of these abnormal bodies as average may be found in normal blood.

The skin biopsies have no use at all, as authors show, therefore they should not be presented. Perhaps they could be mentioned but discussed why they do not show the same appearance as the blood cells.

Blood cells are useful since blood goes all over the body and crossed the brain barrier, but the blood cells to use should be tipified and selected for optical microscopy. 

Figure 3 is difficult to follow and useless. The results are better summarized in the text.

Altogether, authors are encouraged to improve the study using a type of blood cell, with optic microscopy, and using appropriate controls, and showing many cells in a field.

Reviewer 3 Report

The authors have used electron microscopy technique to observe abnormal storage inclusions in peripheral blood buffy coat. After ERT there is reduction in the storage inclusions. It would be ideal if the authors could include electron micrograph images with same magnification for the images after ERT. 3000x images of samples after ERT treatment would be ideal.